# Association between lignan polyphenol bioavailability and enterotypes of isoflavone metabolism: A cross-sectional analysis

**Tomoko Fujitani[1], Mariko Harada Sassa[1], Zhaoqing Lyu[1], Yukiko Fujii[2], Kouji H. Harada[1]***

**1** Department of Health and Environmental Sciences, Kyoto University Graduate School of Medicine, Kyoto, Japan, **2** Department of Pharmaceutical Sciences, Daiichi University of Pharmacy, Fukuoka, Japan

* kharada-hes@umin.ac.jp

**Data Availability Statement:** The data that support the findings of this study are available from the corresponding author upon reasonable request. This is due to the ethical restrictions under the

## Abstract

Lignan polyphenols derived from plants are metabolized by bacteria in the gut to mammalian lignans, such as enterolactone (ENL) and enterodiol (END). Mammalian lignan intake has been reported to be associated with obesity and low blood glucose levels. However, the factors that are responsible for individual differences in the metabolic capacity for ENL and END are not well understood. In the present study, the effects of enterotypes of isoflavone metabolism, equol producers (EQP) and O-desmethylangolensin producers (O-DMAP), on lignan metabolism were examined. EQP was defined by urinary daidzein (DAI) and equol concentrations as $\log(\text{equol/DAI}) \geq -1.42$. O-DMAP was defined by urinary DAI and O-DMA concentrations as O-DMA/DAI > 0.018. Isoflavone and lignan concentrations in urine samples from 440 Japanese women were measured by gas chromatography-mass spectrometry. Metabolic enterotypes were determined from the urinary equol and O-DMA concentrations. Urinary END and ENL concentrations were compared in four groups, combinations of EQP (+/−) and O-DMAP (+/−). The urinary lignan concentration was significantly higher in the O-DMAP/EQP group (ENL: P<0.001, END: P<0.001), and this association remained significant after adjusting for several background variables (END: β = 0.138, P = 0.00607 for EQP and β = 0.147, P = 0.00328 for O-DMAP; ENL: β = 0.312, P<0.001 for EQP and β = 0.210, P<0.001 for O-DMAP). The ENL/END ratio was also highest in the O-DMAP/EQP group, indicating that equol and O-DMA metabolizing gut bacteria may be involved in lignan metabolism. In conclusion, urinary lignan concentrations were significantly higher in groups containing either EQP or O-DMAP than in the non-EQP/non-O-DMAP group. The variables and participants in this study were limited, which the possibility of confounding by other variables cannot be ruled out. However, there are no established determinants of lignan metabolism to date. Further research is needed to determine what factors should be considered, and to examine in different settings to confirm the external validity.

Ethical Guidelines for Medical and Biological Research Involving Human Subjects by the Ministry of Education, Culture, Sports, Science and Technology and the Ministry of Health, Labour and Welfare of Japan. Inquiries about data access should be sent to: kuhes-office@umin.ac.jp (Office of Kyoto University Human Specimen Bank). Data is maintained both at Kyoto University Human Specimen Bank and by the principal investigator (Kouji H. Harada).

**Funding:** This study was supported by a Grant-in-Aid for Scientific Research from the Japan Society for the Promotion of Science (JSPS) (Grant No. 20H03930). The funders had no role in the design of the study; in the collection, analyses, or interpretation of data; in the writing of the manuscript; or in the decision to publish the results.

**Competing interests:** The authors have declared that no competing interests exist.

## 1. Introduction

Phytoestrogens are found in plant foods, such as soybeans, nuts, flax seeds, and vegetables and fruits, and when ingested, foods containing phytoestrogens are metabolized by intestinal bacteria, absorbed, and transported throughout the body via the bloodstream. Each substance has a specific metabolic pathway, which requires the presence of bacteria. Because the gut microbiota varies from person to person, there are individual differences in metabolic capacity [1]. Phytoestrogens can be broadly divided into flavonoids and non-flavonoids.

Lignan polyphenols are non-flavonoids that have been reported to have anti-inflammatory, anticancer, and cardiovascular disease risk-reducing effects [2]. Lignan polyphenols are found in many foods, including fruits and vegetables, such as cucumbers, broccoli, and carrots, and wine and tea [3]. Representative mammalian lignans, enterodiol (END) and enterolactone (ENL) are metabolized from secoisolariciresinol (SECO) [4]. It has been reported that there were negative relationships between high urinary ENL/END and obesity, C-reactive protein levels, and blood glucose levels [5]. There are differences in the metabolic capacity of ENL and END between individuals, but the factors that are responsible for these differences are not well understood.

Isoflavones, a type of flavonoid, are found in soybeans, and the major isoflavone in soybeans and soy products is daidzein (DAI). DAI is metabolized by intestinal bacteria to produce O-desmethylangolensin (O-DMA) and equol. Equol has been reported to have strong estrogenic effects and may reduce the risk of breast cancer, hot flashes, coronary heart disease, and dementia [6]. People who can produce equol are called equol producers (EQP), and the number of EQP in the population is approximately 30%–50% [7–10], which may vary by region and dietary habit. Similarly, people who can produce O-DMA are termed O-DMA producers (O-DMAP) and approximately 80%–90% of the population are O-DMAP. Some studies have reported higher bone mineral density in postmenopausal women who are non-O-DMAP compared with those who are O-DMAP [11]. Healthy postmenopausal Asian women who were O-DMAP were reported to have lower values for body mass index (BMI), body fat percentage and systolic blood pressure, and total cholesterol and low-density lipoprotein (LDL) cholesterol levels than non-O-DMAP [12]. In a study in USA, the study conducted soy challenge for adult males and females, and found an increased risk of obesity in non-O-DMAP compared with O-DMAP, and no association was observed with obesity between EQP and non-EQP, even after adjustment for age, gender, menstrual status, and race [13]. In contrast, a study of Japanese subjects reported that non-EQP comprised 67.9% of the overweight or obese subjects while general population 50% [14].

As mentioned above, to produce equol and O-DMA, it is important not only to consume DAI from soybeans and soy-derived foods, but also to possess the necessary intestinal bacteria. Similarly, the production of END and ENL also requires the intake of foods containing lignan and the presence of particular intestinal bacteria. For example, *Eubacterium ramulus* wK1, strain SY8519 and *Clostridium* spp. HGHA136 gut bacteria are involved in the metabolism of O-DMA [15]. *Eubacterium* sp. ARC-1 has been proposed to be a lignan-metabolizing bacteria [15]. The genus *Eubacterium* may exhibit a common metabolic capacity for lignans and isoflavones.

Our previous preliminary studies on EQP suggested a positive correlation between urinary equol, END, and ENL levels, and that EQP likely harbor gut bacteria that can metabolize lignan [16]. It has also been reported that there was no significant correlation between EQP and O-DMAP [13], and that the metabolic capacity of equol and that of O-DMA were separate and independent. In the present study, we examined individual differences in urinary END and

ENL concentrations for enterotypes of EQP and non-EQP, each of which was further divided into O-DMAP and non-O-DMAP (four enterotypes in total).

## 2. Materials and methods

### 2.1. Ethics statement

The protocol for this study was approved by the Ethics Committee of Kyoto University Graduate School of Medicine and Faculty of Medicine and Hospital (Latest approval number R1478-9 on Aug 12th, 2021, 'Human exposure monitoring and risk assessment'). Written informed consent was obtained from all participants before sample collection.

### 2.2. Study participants

For this study, 440 non-pregnant women from five prefectures recruited in health checkups between November 2000 and December 2001 were randomly selected from the Kyoto Human Specimen Bank as a cross-sectional study. Adult women were included. This was a subpopulation of previous studies [16, 17]. Urine samples were collected at a time of no menstruation and were stored at − 30˚C.

Age and BMI of participants were obtained from health records. The participants' parity, smoking habits, disease histories, and menstrual status (regular cycles, irregular cycles, menopause, experienced gynecological surgery) were obtained using a self-report questionnaire. A regular cycle was defined as a menstrual cycle of 25–38 days. An irregular cycle was defined as menstrual cycles that were not regular (i.e., not 25–38 days) in the previous 3 months. Menopause was defined as no menstruation for 12 months. Those who experienced gynecological surgery were categorized to eliminate the potential effects of the surgery on the intestinal microbiota. All participants were of Asian ethnicity.

### 2.3. Measurement of urinary isoflavones and lignans

Target compounds in the urine were analyzed using gas chromatography-mass spectrometry (6890GC/5973MSD; Agilent Technologies Japan Ltd., Tokyo, Japan) in 2009. Urine (0.5 mL) was incubated with 2,500 U of a glucuronidase/sulfatase solution for 12 h at 37˚C. The solution was then loaded onto a Sep-Pak Plus C18 solid-phase column (Waters, Milford, MA, USA) and eluted with 3 mL of a 1:1 acetonitrile: ethyl acetate mixture. Dried residues were derivatized with 50 μL of N,O-bis(trimethylsilyl)trifluoroacetamide with 1% trimethylchlorosilane at 60˚C for 1 h. Derivatized solutions (1 μL) were injected and separated on a DB-5MS capillary column (length 30 m, inner diameter 0.25 mm, film thickness 0.25 μm), and quantified in electron impact ionization mode. Two lignans (END and ENL) and three isoflavones (DAI, O-DMA, and equol) were analyzed. The detection limits for the analytes were 0.1, 0.5, 20, 16, and 28 ng/mL for ENL, END, equol, O-DMA, and DAI, respectively. Quantification was conducted with an internal standard (d4-daidzein) and nine-point calibration curves (0.1 to 1000 ng/mL). The linearity of the calibration curves (r) ranged from 0.991 (O-DMA) to 0.998 (DAI). The urinary concentrations of the lignans and isoflavones were normalized to urinary creatinine concentrations, grams of creatinine (g-Cr). Creatinine was assayed using an enzymatic method. Changes over time in the target compounds during storage were not evaluated in this study, but in our previous study [17], the archived samples used from the same specimen bank showed similar average concentrations of isoflavones and lignans to fresh samples. EQP were defined as log(equol/DAI) $\geq$ −1.42 [18]. O-DMAP was defined as O-DMA/DAI > 0.018 [19].

## 2.4. Statistical analysis

The primary outcome is the relationship between four enterotypes and urinary lignans concentrations.

To enable statistical analysis, the urinary concentrations were transformed into common logarithmic values because the concentrations were not normally distributed. Values under the detection limits were recorded as half of the detection limit for statistical analyses. Differences in the means and proportions were examined using ANOVA and Fisher's exact test, respectively (two-tailed). To adjust the potential confounders, we conducted analyses of covariance. Records with missing values were planned to be excluded from the analyses while there were no missing values in the data analyzed. JMP Pro (ver. 16; SAS Institute, Cary, NC, USA) was used to perform these calculations. The alpha level for all tests was set at 0.05. Sample size was determined based on laboratory analytical capabilities.

# 3. Results

## 3.1. Characteristics of study subjects

Table 1 shows the characteristics of the study subjects and the urinary lignan and isoflavone levels. The average age of the participants was 49.2 years (range: 23–74 years). Approximately half of the participants reported having regular menstrual cycles, and 11.7% were smokers. The numbers of participants that reported a current disease and/or a history of disease were 9 (current disease: 1) for myoma of the uterus; 21 (10) thyroid disease; 5 (5) hypertension; 5 (2) cancer; 8 (0) liver disease; 15 (4) kidney disease; 4 (0) bile stone; and 50 (13) other diseases. In addition, 20 of the participants had multiple diseases: 16 had two diseases and 4 had three diseases.

Urinary ENL and O-DMA concentrations differed significantly according to menstrual status: the urinary ENL level was higher in women with an irregular cycle than with a regular cycle (P = 0.0344) and urinary O-DMA levels were higher in the menopause group than in women with a regular cycle (P = 0.0112). Because there were significant differences in the ENL and O-DMA levels for women with different menstrual status, the correlation coefficients between age and ENL and O-DMA were examined, and no correlation was found for ENL (r = 0.0417, P = 0.381) and only a weak correlation was shown for O-DMA (r = 0.136, P = 0.00419). Therefore, we considered the effects of menstruation and age on ENL and O-DMA levels to be small.

Overall, 48.9% of the study participants were classified as EQP and 79.5% as O-DMAP. A χ2 test between the EQP and O-DMAP enterotypes revealed a significant difference in association between EQP and O-DMAP (χ2 test: P < 0.0001), with an odds ratio of 2.59 for O-DMAP to EQP (S1 Table).

The association of EQP and O-DMAP with urinary lignan polyphenols and iso-flavones, respectively, was examined. Urinary END levels were higher in EQP and O-DMAP than in non-EQP and non-O-DMAP, respectively (Table 1, EQP: P = 0.000305, O-DMAP: P < 0.0001). Urinary ENL levels were also higher in EQP and O-DMAP than in non-EQP and non-O-DMAP, respectively (Table 1, EQP: P < 0.0001, O-DMAP: P < 0.0001). The levels of urinary DAI, as a representative urinary isoflavone, were lower for EQP and O-DMAP than for non-EQP and non-O-DMAP, respectively (Table 1, EQP: P < 0.0001, O-DMAP: P < 0.0001). Urinary O-DMA levels were lower in EQP than in non-EQP. Urinary equol was higher in non-O-DMAP than in EQP (O-DMAP: P = 0.00251).

**Table 1. Characteristics of the study subjects and the urinary lignan and isoflavone concentrations.**

| | | urinary lignan and isoflavone (μmol/g-Cr), GM (GSD) | | | | |
|---|---|---|---|---|---|---|
| | | Lignan | | Isoflavone | | |
| | | END | ENL | DAI | O-DMA | Equol |
| Total | N = 444 | 0.124 (6.09) | 0.0818 (7.13) | 6.14 (4.56) | 0.493 (10.3) | 0.313 (6.43) |
| Age (yr, mean±SD) | 49.29± 10.11 | | | | | |
| Menstrual cycle (n [%]) | | | | | | |
| Regular cycles | 204 (45.9%) | 0.114 (6.46) | 0.0675 (7.22) | 5.01 (4.24) | 0.351 (9.17) | 0.245 (6.54) |
| Irregular cycles | 39 (8.8%) | 0.154 (5.07) | 0.165 (6.27) | 6.33 (4.51) | 0.556 (10.4) | 0.464 (7.50) |
| Menopause | 158 (35.6%) | 0.140 (6.18) | 0.0935 (7.17) | 7.41 (4.95) | 0.790 (10.9) | 0.393 (6.01) |
| Experienced gynecological surgery | 33 (7.4%) | 0.0889 (5.65) | 0.0580 (6.73) | 8.10 (4.59) | 0.525 (9.72) | 0.317 (5.78) |
| Smoking habit (n [%]) | | | | | | |
| Non- smoker | 363 (81.8%) | 0.133 (6.13) | 0.0879 (7.24) | 6.16 (4.63) | 0.509 (10.7) | 0.344 (6.44) |
| Current smoker | 52 (11.7%) | 0.0871 (5.80) | 0.0523 (6.98) | 5.95 (4.01) | 0.49 (7.76) | 0.161 (5.57) |
| Ex- smoker | 12 (2.7%) | 0.151 (3.62) | 0.0926 (6.61) | 2.76 (5.04) | 0.385 (7.46) | 0.331 (8.36) |
| BMI (n [%]) | | | | | | |
| Underweight | 73 (16.4%) | 0.130 (5.89) | 0.0894 (6.70) | 6.70 (4.44) | 0.577 (9.14) | 0.480 (7.21) |
| Normal | 271 (61.0%) | 0.121 (6.35) | 0.0816 (7.64) | 6.12 (4.72) | 0.481 (10.8) | 0.276 (6.51) |
| Overweight | 92 (20.7%) | 0.133 (5.95) | 0.0754 (6.14) | 5.73 (4.44) | 0.458 (9.95) | 0.32 (5.70) |
| Obesity | 8 (1.8%) | 0.0942 (2.21) | 0.0983 (4.97) | 7.29 (2.94) | 0.621 (11.4) | 0.363 (3.93) |
| Disease history (n [%]) | | | | | | |
| None | 321 (72.3%) | 0.130 (6.13) | 0.0844 (7.11) | 5.78 (4.43) | 0.482 (10.4) | 0.327 (6.61) |
| Current/past histories | 123 (27.7%) | 0.110 (5.98) | 0.0753 (7.20) | 7.22 (4.91) | 0.523 (10.0) | 0.280 (6.00) |
| EQP (n [%]) | | | | | | |
| EQP | 217 (48.9%) | 0.170 (5.78) | 0.166 (5.83) | 2.82 (4.84) | 0.326 (8.68) | 1.03 (6.79) |
| Non EQP | 227 (51.1%) | 0.0919 (6.10) | 0.0416 (6.75) | 12.9 (2.72) | 0.733 (11.2) | 0.0999 (2.18) |
| O-DMAP (n [%]) | | | | | | |
| O-DMAP | 353 (79.5%) | 0.148 (5.92) | 0.106 (6.65) | 5.34 (4.56) | 1.02 (6.70) | 0.358 (6.82) |
| Non O-DMAP | 91 (20.5%) | 0.0625 (5.78) | 0.0295 (6.72) | 10.6 (4.08) | 0.0290 (4.63) | 0.185 (4.55) |

Values of urinary ENL, END, DAI, O-DMA, and equol were geometric means (GMs) and standard deviations (GSDs). END: enterodiol; ENL: enterolactone; O-DMA: O-desmethylangolensin; DAI: daidzein; Log: common logarithm; gCr: grams creatinine; CI: confidence interval; O-DMAP: O-DMA producer; EQP: equol producer; BMI: body mass index.

### 3.2. Urinary lignan excretion of the four isoflavone metabolism enterotypes

Isoflavone metabolism was divided into four enterotypes, initially by classification as: EQP and non-EQP, which were then further classified as O-DMAP and non-O-DMAP. The urinary END and ENL concentrations for the four enterotypes were analyzed using one-way analysis of variance (Fig 1). There was up to a 10-fold difference in urinary lignan concentrations between the enterotypes. The geometric mean of the urinary log END (0.0473 μmol/g-Cr) value in non-O-DMAP/non-EQP was lower than for O-DMAP/EQP and O-DMAP/non-EQP (0.180 and 0.119 μmol/g-Cr; P < 0.001 and P = 0.0025, respectively). Also, urinary log ENL (0.0176 μmol/g-Cr) concentrations were lower in non-EQP/non-O-DMAP than in EQP/O-DMAP, non-EQP/O-DMAP, and non-O-DMAP/EQP (0.181, 0.0577 and 0.0943 μmol/g-Cr; P < 0.001, P < 0.001, and P = 0.0003, respectively).

Urinary equol and O-DMA are produced by the metabolization of DAI, therefore, increased urinary lignan levels might be confounded with the dietary intake of isoflavones. To test this possibility, we examined the correlation between levels of urinary DAI and levels of urinary END and ENL in EQP and non-EQP (Fig 2). A weak correlation between log DAI

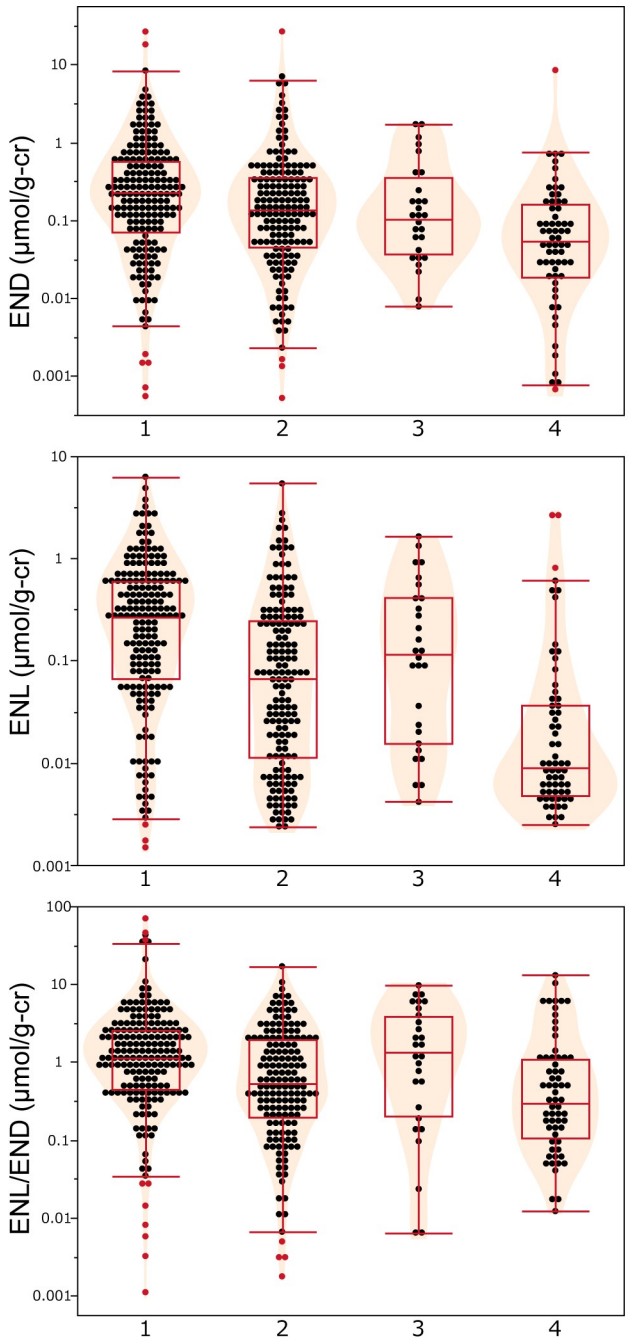

1 O-DMAP/EQP, 2 O-DMAP/non-EQP, 3 Non-O-DMAP/EQP, 4 non-O-DMAP/non-EQP

**Fig 1. Urinary mammalian lignan concentrations and ratios between isoflavone metabolism enterotypes.** END: enterodiol, ENL: enterolactone, END/ENL: ratio of END to ENL, O-DMAP: O-DMA producer; EQP: equol producer. The boxes show the interquartile ranges, the center lines in the boxes show the medians, and the upper and lower whiskers show the 95th and 5th percentiles. The width of the violin indicates the probability density.

(μmol/g-Cr) and log END (μmol/g-Cr) was demonstrated in EQP (r = 0.167, P = 0.0138), but was not significant in non-EQP (r = 0.0169, P = 0.800) (Fig 2). There was also a weak correlation observed between the log DAI and log ENL values (r = 0.192, P = 0.00443) for EQP, but not for non-EQP (r = 0.0511, P = 0.443) (Fig 2). Furthermore, these relationships were

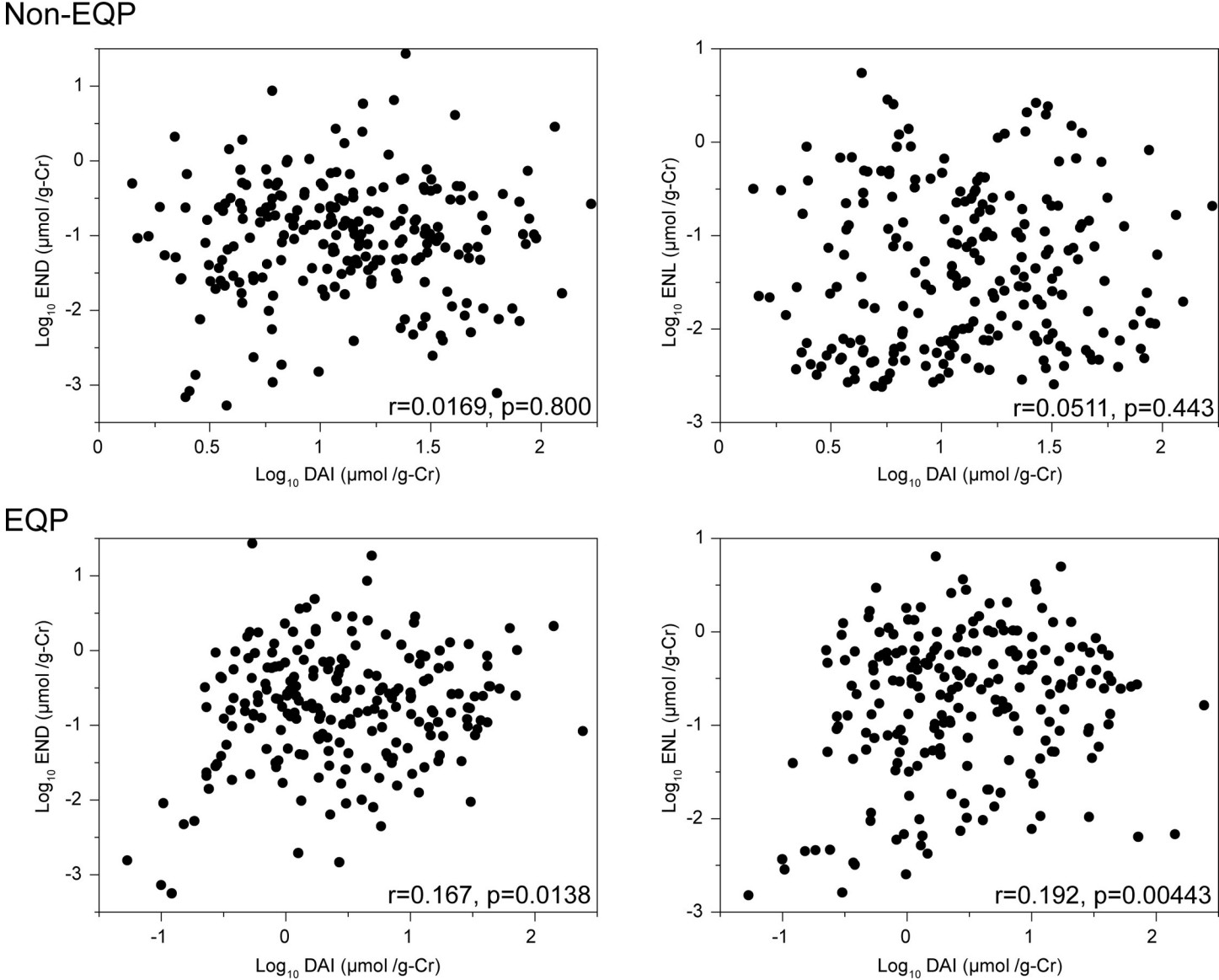

**Fig 2. Scatter plots of the relationship between urinary daidzein (DAI) and mammalian lignans.** Urinary concentrations of DAI, enterodiol (END), and enterolactone (ENL) were common log-transformed. The correlations were assessed by equol producer (EQP)enterotypes. Pearson's correlation coefficients were examined.

examined in the four enterotypes (S2 Table). Only in O-DMAP/EQP weak correlations were observed (END: r = 0.244, P = 0.000704; ENL: r = 0.254, P = 0.000427) (S2 Table). According to these results, the relationship between the four enterotypes and the urinary END and ENL concentrations was not likely to be affected by soy consumption.

The ENL/END ratio was further examined as an index of the metabolism of END to ENL. This index is not likely to be affected by soy isoflavone intake. Among the four enterotypes, the O-DMAP/EQP group showed the highest ENL/END ratio (ANOVA: P<0.05, Fig 1). The EQP/non-O-DMAP group also had a higher ENL/END ratio than the non-O-DMAP/non-EQP group, suggesting that EQP have a greater ability to metabolize END to ENL.

### 3.3. Multivariable regression analysis of the associations between isoflavone metabolism enterotypes and urinary lignan excretion

The effects of isoflavone metabolism enterotypes on urinary END and ENL concentrations were further examined in a multivariable analysis using age, menstrual status, smoking habit, BMI, disease histories, EQP, O-DMAP, and an interaction term between EQP and O-DMAP as independent variables (Table 2). For log urinary END (μmol/g-Cr) values, EQP and O-DMAP were significant factors (β = 0.138, P < 0.00607 and β = 0.147, P = 0.00328, respectively). EQP and O-DMAP were also significantly associated with log urinary ENL (μmol/g-Cr) values (β = 0.312, P < 0.0001 and β = 0.210, P < 0.0001, respectively). The interaction term between EQP and O-DMAP was not significant for log urinary END (μmol/g-Cr) and log urinary ENL (μmol/g-Cr) values (END: β = -0.0474, P = 0.343; ENL: β = -0.0598, P = 0.240). These results indicated that EQP and O-DMAP independently affected the metabolism of END and ENL.

Because of the possible influence of soy isoflavone intake on the definitions of EQP and O-DMAP, we performed the sensitivity analysis in samples with urinary DAI concentrations greater than the 25th percentile of the total samples (DAI ≥ 575 μg/g-Cr) (Table 3). Log urinary END (μmol/g-Cr) was associated marginally with EQP (β = 0.110, P = 0.0563) and significantly associated with O-DMAP (β = 0.221, P < 0.0001). For log urinary ENL (μmol/g-Cr), both EQP and O-DMAP were significantly associated (β = 0.354, P < 0.0001 and β = 0.257, P < 0.0001, respectively). The interaction terms were not significant. Among the adjustment factors, irregular cycles showed an association with urinary ENL (β = 0.244, P = 0.0461). The biological relevance of this relationship is unknown. It is possible that the estrogenic effects of lignans may affect the menstrual status, but a previous study has demonstrated that there was no association between menstrual cycles and lignan polyphenols [20].

## 4. Discussion

There have not been reported for determinants of availability of lignan polyphenols while inter-individual variability was found. In the present study, we examined individual differences in urinary END and ENL concentrations for enterotypes of isoflavone metabolisms. To date, there have been no studies examining the relationship between lignans and the type of metabolism of soy isoflavones by enteric bacteria. In this observational study, the metabolic potential of END and ENL was examined in four enterotypes, classified by EQP and O-DMAP, and the urinary ENL concentration in the non-EQP/non-O-DMAP group was the lowest of the four enterotypes. The results suggested that the ingested plant lignan polyphenols were poorly utilized in the non-EQP/non-O-DMAP group. The urinary END level was also affected by the enterotype. Furthermore, the ratio of END to ENL metabolism was higher in the EQP group. Thus, isoflavone metabolism enterotypes were shown to be one of the factors responsible for the interindividual differences in lignan metabolism.

Several different intestinal bacteria have been reported to determine whether individuals are O-DMAP and EQP. In the present study, O-DMAP was associated with EQP (S1 Table), suggesting that EQP and O-DMAP may have gut bacteria in common. Both equol and O-DMA are produced from dihydrodaidzein. *Coprobacillus* sp. strain TM-40 metabolizes dihydrodaidzein from DAI [21]. Lactic acid bacteria and *Bifidobacterium* strains have also been found to produce O-DMA and dihydrodaidzein [22]. These bacteria may be shared in O-DMAP and EQP. In contrast, the gut bacteria, such as *Slackia equolifaciens*, that ultimately metabolize dihydrodaidzein to equol are likely to be different in O-DMAP compared with EQP [23].

**Table 2. Association between the log urinary lignan value and EQP and O-DMAP enterotypes with adjusted variables.**

| Dependent variable: Log urinary END (μmol/g-Cr) (n = 420) | | | |
|---|---|---|---|
| Independent variables | β | 95% CI | P |
| Age (yr) | -0.000856 | -0.0147–0.0129 | 0.903 |
| Menstrual status (irregular cycles) | 0.0640 | -0.146–0.274 | 0.549 |
| (menopause) | 0.0560 | -0.113–0.225 | 0.516 |
| (experienced gynecological surgeries) | -0.115 | -0.371–0.141 | 0.377 |
| EQP | 0.138 | 0.0397–0.237 | 0.00607 |
| O-DMAP | 0.147 | 0.0494–0.245 | 0.00328 |
| EQP*O-DMAP | -0.0474 | -0.146–0.0507 | 0.343 |
| Smoking habit (current smoker) | -0.0837 | -0.293–0.125 | 0.432 |
| (ex-smoker) | 0.0340 | -0.272–0.340 | 0.827 |
| BMI (normal) | -0.0139 | -0.116–0.0882 | 0.789 |
| (overweight and obese) | 0.0194 | -0.106–0.145 | 0.762 |
| Disease histories (current or past) | -0.00301 | -0.0984–0.0924 | 0.951 |
| Dependent variable: Log urinary ENL (μmol/g-Cr) (n = 420) | | | |
| Independent variables | β | 95% CI | P |
| Age (yr) | 0.000594 | -0.0135–0.0147 | 0.934 |
| Menstrual status(irregular cycles) | 0.208 | -0.00544–0.422 | 0.0561 |
| (menopause) | -0.00936 | -0.182–0.163 | 0.915 |
| (experienced gynecological surgeries) | -0.133 | -0.394–0.127 | 0.315 |
| EQP | 0.312 | 0.212–0.413 | **<0.0001** |
| O-DMAP | 0.210 | 0.111–0.310 | **<0.0001** |
| EQP*O-DMAP | -0.0598 | -0.160–0.0402 | 0.240 |
| Smoking habit (current smoker) | -0.0164 | -0.230–0.197 | 0.880 |
| (ex-smoker) | -0.0614 | -0.373–0.250 | 0.699 |
| BMI (normal) | 0.00445 | -0.0997–0.109 | 0.933 |
| (overweight and obese) | -0.0246 | -0.152–0.103 | 0.706 |
| Disease histories (current or past) | -0.00404 | -0.101–0.0932 | 0.935 |

END: enterodiol; ENL: enterolactone; Log: common logarithm; gCr: grams of creatinine; CI: confidence interval; O-DMAP: O-DMA producer; EQP: equol producer; EQP*O-DMAP: the interaction term between EQP and O-DMAP; BMI: body mass index. EQP status was coded as 1 = EQP and 0 = non-EQP as reference; O-DMAP status was coded as 1 = O-DMAP and 0 = non-O-DMAP as reference; tobacco use was coded as 1 = non-smoker as reference, 2 = current smoker, and 3 = ex-smoker; menstrual status was coded as 1 = regular cycles as reference, 2 = irregular cycles, 3 = menopause, and 4 = experienced gynecological surgery; BMI was coded as 1 = underweight as reference, 2 = normal weight, and 3 = overweight or obese; disease history was coded as 0 = none as reference and 1 = current or past. Model fitness: $R^2$ = 0.0663 for log urinary END; $R^2$ = 0.192 for log urinary ENL.

Bacteria that metabolize lignans have been reported previously. Two bacterial strains, *Peptostreptococcus* sp. SDG-1 and *Eubacterium* sp. SDG-2, were shown to be responsible for the transformation of SECO to the mammalian lignan END in an isolated human fecal suspension [24]. Strains of *Bacteroides distasonis*, B. *fragilis*, B. *ovatus*, *Clostridium cocleatum*, and *Clostridium* sp. SDG-Mt85-3Db deglycosylated SECO glycosides to SECO. In the next step, demethylation of SECO was catalyzed by strains of *Butyribacterium methylotrophicum*, *E. callanderi*, *E. limosum*, and *Peptostreptococcus productus*. Dehydroxylation of SECO was catalyzed by strains of *C. scindens* and *Eggerthella lenta*. Dehydrogenation of END to ENL has been reported to be catalyzed by the strain ED-Mt61/PYG-s6 [25]. Further research is needed to determine what bacteria are involved in both isoflavone and lignan metabolism.

The preventive effects of polyphenols have been studied extensively. Among them, equal has higher estrogenic activity than other isoflavones, and there are several reports of a reduced

**Table 3. Association between log urinary lignan and enterotypes among samples with moderate-to-high daidzein concentration.**

| Dependent Variable: Log urinary END (µmol/g-Cr) (n = 312) | | | |
|---|---|---|---|
| Independent variables | β | 95% CI | P |
| Age (yr) | -0.00384 | -0.0197–0.0120 | 0.633 |
| Menstrual status (irregular cycles) | 0.0207 | -0.218–0.260 | 0.865 |
| (menopause) | 0.0958 | -0.0919–0.284 | 0.316 |
| (experienced gynecological surgeries) | -0.108 | -0.399–0.183 | 0.467 |
| EQP | 0.110 | -0.00297–0.222 | 0.0563 |
| O-DMAP | 0.221 | 0.108–0.334 | < 0.0001 |
| EQP*O-DMAP | 0.0314 | -0.0819–0.145 | 0.586 |
| Smoking habit (current smoker) | -0.170 | -0.419–0.0787 | 0.179 |
| (ex-smoker) | 0.222 | -0.164–0.608 | 0.259 |
| BMI (normal) | -0.0322 | -0.147–0.0830 | 0.583 |
| (overweight and obese) | -0.00553 | -0.147–0.136 | 0.939 |
| Disease histories (current or past) | 0.00461 | -0.103–0.112 | 0.933 |
| Dependent Variable: Log urinary ENL (µmol/g-Cr) (n = 312) | | | |
| Independent variables | β | 95% CI | P |
| Age (yr) | -0.00116 | -0.0171–0.0147 | 0.886 |
| Menstrual status (irregular cycles) | 0.244 | 0.00425–0.484 | 0.0461 |
| (menopause) | 0.0310 | -0.1580–0.219 | 0.749 |
| (experienced gynecological surgeries) | -0.241 | -0.533–0.0513 | 0.106 |
| EQP | 0.354 | 0.241–0.467 | < 0.0001 |
| O-DMAP | 0.257 | 0.144–0.371 | < 0.0001 |
| EQP*O-DMAP | -0.0147 | -0.129–0.0992 | 0.800 |
| Smoking habit (current smoker) | -0.0214 | -0.272–0.229 | 0.866 |
| (ex-smoker) | 0.00957 | -0.379–0.398 | 0.961 |
| BMI (normal) | 0.0174 | -0.0984–0.133 | 0.768 |
| (overweight and obese) | -0.0442 | -0.186–0.0980 | 0.541 |
| Disease histories (current or past) | -0.00946 | -0.117–0.0982 | 0.863 |

END: enterodiol; ENL: enterolactone; Log: common logarithm; gCr: grams creatinine; CI: confidence interval; O-DMAP: O-DMA producer; EQP: equol producer; EQP*O-DMAP: the interaction term between EQP and O-DMAP; BMI: body mass index. EQP status was coded as 1 = EQP and 0 = non-EQP as reference; O-DMAP status was coded as 1 = O-DMAP and 0 = non-O-DMAP as reference; tobacco use was coded as 1 = non-smoker as reference, 2 = current smoker, and 3 = ex-smoker. Menstrual status was coded as 1 = regular cycles as reference, 2 = irregular cycles, 3 = menopause, and 4 = experienced gynecological surgery; BMI was coded as 1 = underweight, 2 = normal weight, and 3 = overweight or obese; Disease history was coded as 0 = none as reference and 1 = current or past. Model fitness: $R^2 = 0.0105$ for log urinary END; $R^2 = 0.256$ for log urinary ENL.

risk of hormone-derived cancers, such as breast cancer, and reduced hot flashes, with high levels of equol [26]. However, there have been few reports on the effects of O-DMA on health, and its usefulness is unclear. For lignans, a correlation between blood ENL levels and a lower incidence of coronary heart disease and lower mortality has been reported [27]. The present study suggested that lignan availability may be modified by EQP and O-DMAP, and the effects of equol and O-DMAP observed in previous studies may include lignan-mediated effects.

A more comprehensive evaluation of polyphenols in epidemiological studies is desirable. Blood ENL concentrations have been reported to be weakly correlated with lignan intake as assessed by FFQ [28]. Because individual differences in gut microbiota are not evaluated, the estimation of mammalian lignans by FFQ is considered to have a large uncertainty. The study population in future epidemiological and intervention studies should be optimized to characterize the metabolism of lignan polyphenols using O-DMAP and EQP status as enterotype

indicators. Furthermore, metagenomic analysis of the gut microbiota is expected to identify bacteria that determine enterotypes, more accurately and precisely assess lignan utilization, and develop interventions to improve the gut microbiota.

This study determined the metabolic enterotype of isoflavone metabolism from urinary equol and O-DMA concentrations but did not directly measure the intestinal microbiota. In the future, a metagenomic analysis of the bacterial flora will be performed to identify the specific bacteria involved in the metabolism. However, it is an advantage of this study that enterotypes could be easily determined from the urinary isoflavones. Because there were no data on isoflavone and lignan intake for individuals, urinary isoflavone and lignan metabolite concentrations were used to categorize enterotypes. In individuals with low isoflavone intakes, this categorization may lead to misclassification and possible confounding of the enterotypes. However, because the DAI concentrations were not strongly associated with lignan concentrations (Fig 2), it was unlikely that major confounding or misclassification of enterotypes occurred. In addition, a significantly higher ratio of ENL/END was observed in the EQP/O-DMAP group, suggesting that the enterotype may be a determinant of END metabolism. The variables used in the multiple regression analysis, including age, BMI, and menstrual status, were limited. The possibility of confounding by other variables cannot be ruled out. However, because there are no established determinants of lignan metabolism to date, further research is needed to determine what factors should be considered. Finally, the participants in this study were adult women of Asian ethnicity living in Japan. The association should be examined in different settings to confirm the external validity.

## 5. Conclusion

In this study, the factors that are responsible for individual differences in the bioavailability of ENL and END were investigated in relation to enterotypes of isoflavone metabolism. Urinary END and ENL concentrations were 3.80- and 10.2-fold higher in the EQP/O-DMAP group than in the non-EQP/non-O-DMAP group, respectively. However urinary isoflavone concentrations were not associated with END and ENL concentrations. Dietary intake of lignan polyphenols may not represent precise bioavailable amounts and these enterotypes may provide better estimates. The possibility of biases and confounding cannot be ruled out due to the limitations in observed variables and participants. Further research to determine causative gut bacteria is needed.

## Supporting information

**S1 Table. The ratio of EQP enterotype to O-DMAP enterotype.**
(DOCX)

**S2 Table. Pearson's correlation between urinary log DAI values and urinary log ENL and log END values by EQP enterotype and O-DMAP enterotype.**
(DOCX)

## Acknowledgments

The authors would like to express their sincere appreciation to Prof. Dr. Akio Koizumi, Professor Emeritus of Kyoto University, founder of the Kyoto University Human Specimen Bank, and also to our many collaborators. Victoria Muir, PhD, from Edanz (https://jp.edanz.com/ac) edited a draft of this manuscript.

## Author Contributions

**Conceptualization:** Tomoko Fujitani, Mariko Harada Sassa, Kouji H. Harada.

**Formal analysis:** Tomoko Fujitani.

**Funding acquisition:** Kouji H. Harada.

**Investigation:** Zhaoqing Lyu, Yukiko Fujii, Kouji H. Harada.

**Project administration:** Kouji H. Harada.

**Supervision:** Kouji H. Harada.

**Writing – original draft:** Tomoko Fujitani.

**Writing – review & editing:** Mariko Harada Sassa, Zhaoqing Lyu, Yukiko Fujii, Kouji H. Harada.

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
