## [Decision Letter · Decision Letter 0]

11 Sep 2023

PONE-D-23-19702Lignan polyphenol metabolism is partially determined by isoflavone metabolism enterotypes: a cross-sectional analysisPLOS ONE

Dear Dr. Harada,

Thank you for submitting your manuscript to PLOS ONE. After careful consideration, we feel that it has merit but does not fully meet PLOS ONE’s publication criteria as it currently stands. Therefore, we invite you to submit a revised version of the manuscript that addresses the points raised during the review process.

We look forward to receiving your revised manuscript.

Kind regards,

Awatif Abid Al-Judaibi, PhD

Academic Editor

PLOS ONE

Reviewers' comments:

Reviewer's Responses to Questions

**Comments to the Author**

1. Is the manuscript technically sound, and do the data support the conclusions?

Reviewer #1: Yes

Reviewer #2: No

2. Has the statistical analysis been performed appropriately and rigorously? 

Reviewer #1: Yes

Reviewer #2: No

3. Have the authors made all data underlying the findings in their manuscript fully available?

Reviewer #1: No

Reviewer #2: Yes

4. Is the manuscript presented in an intelligible fashion and written in standard English?

Reviewer #1: Yes

Reviewer #2: No

5. Review Comments to the Author

Reviewer #1: The manuscript aims at studying of lignan metabolism using modern analytical methods. I have the following questions, comments and remarks:

Comments and suggestions:

• Abstract. The experimental groups should be defined.

• Experimental procedures/ Results and Discussion. Did the authors monitor the changes of the detected compounds in time, i.e. the time of storage?

Conclusion

I would suggest to response minor issues prior to further consideration of the work. Based on my concerns, minor revision is required.

Reviewer #2: I have reviewed the manuscript titled Lignan polyphenol metabolism is partially determined by isoflavone metabolism enterotypes: a cross-sectional analysis, and I appreciate the authors' efforts in conducting this research. However, after careful evaluation, I regret to inform you that I cannot recommend this manuscript for publication in its current form due to “Lack of Significance”, “Weak Methodology”, “Less Clarity in Data Presentation”, and “Less Statistical Significance”. I encourage the authors to consider revising the manuscript, addressing the aforementioned concerns, and potentially exploring alternative avenues for publication.

6. PLOS authors have the option to publish the peer review history of their article (what does this mean?). If published, this will include your full peer review and any attached files.

Reviewer #1: No

Reviewer #2: No

---

## [Author Response · Author response to Decision Letter 0]

18 Oct 2023

Responses for reviewers’ comments

Dear Editor and reviewers,

Thank you for valuable feedbacks to our manuscript. We revised our manuscript based on your suggestions. Please see following responses and revisions.

In addition, according to the editorial instructions, ethical statements were moved from Acknowledgments section to Methods section (L74–78, L298–301).

Data availability section was revised to state the reason for the restriction (L306–309).

Reviewer #1: The manuscript aims at studying of lignan metabolism using modern analytical methods. I have the following questions, comments and remarks:

Comments and suggestions:

#1. Abstract. The experimental groups should be defined.

(response)

Thank you for your comments.

We revised the manuscripts as suggested to include the definition of EQP and O-DMAP (L18–20 in changes-tracked manuscript).

EQP was defined by urinary daidzein (DAI) and equol concentrations as log(equol/DAI) ≥   –1.42. O-DMAP was defined by urinary DAI and O-DMA concentrations as O-DMA/DAI > 0.018.

#2. Experimental procedures/ Results and Discussion. Did the authors monitor the changes of the detected compounds in time, i.e. the time of storage?

(response)

Thank you for the valuable comment.

Changes over time of the detected compounds were not measured in this study. But in our previous study, we evaluated the stability of isoflavones and lignans. We revised the manuscripts to include this information (L107–109 in changes-tracked manuscript).

Changes over time in the target compounds during storage were not evaluated in this study, but in our previous study [17], the archived samples used from the same specimen bank showed similar average concentrations of isoflavones and lignans to fresh samples.

#3. Conclusion

I would suggest to response minor issues prior to further consideration of the work. Based on my concerns, minor revision is required.

 Thank you for your evaluation and kind comments.

 

Reviewer #2: I have reviewed the manuscript titled Lignan polyphenol metabolism is partially determined by isoflavone metabolism enterotypes: 

a cross-sectional analysis, and I appreciate the authors' efforts in conducting this research. However, after careful evaluation, I regret to inform you that I cannot recommend this manuscript for publication in its current form due to “Lack of Significance”, “Weak Methodology”, “Less Clarity in Data Presentation”“Less Statistical　Significance”.

I encourage the authors to consider revising the manuscript, addressing the aforementioned concerns, and potentially exploring alternative avenues for publication.

(response)

Thank you for your comments. We revised the manuscript regarding for “Lack of Significance”, “Weak Methodology”,“Less Clarity in Data Presentation”, “Less Statistical Significance”.

“Lack of Significance”

(response)

We emphasized the significance of the study more in the discussion section. (L231–233 in changes-tracked manuscript)

There have not been reported for determinants of availability of lignan polyphenols while inter-individual variability was found. In the present study, we examined individual differences in urinary END and ENL concentrations for enterotypes of isoflavone metabolisms.

“Weak Methodology”

(response)

We have already written the limitation of the study in the Discussion section, but we also added it to the abstract as well. (L28–31 in changes-tracked manuscript)

The variables and participants in this study was limited, which the possibility of confounding by other variables cannot be ruled out. However, there are no established determinants of lignan metabolism to date. Further research is needed to determine what factors should be considered, and to examine in different settings to confirm the external validity.

“Less Clarity in Data Presentation” 

(response)

Since the primary outcome is the relationship between gut microbiota types and urinary lignans, this is clearly indicated in the tables and figures. But we now clearly indicate it in the statistical analysis section (L112 in changes-tracked manuscript). 

The primary outcome is the relationship between four enterotypes and urinary lignans concentrations. 

“Less Statistical　Significance”. 

(response)

The analysis of the primary outcome shows sufficient statistical significances (P<0.01 to P<0.001).

The p-values have been added to the Abstract (L24–26)

The urinary lignan concentration was significantly higher in the O-DMAP/EQP group (ENL: P<0.001, END: P<0.001), and this association remained significant after adjusting for several background variables (END:β=0.138 P=0.00607 for EQP, β=0.147 P= 0.00328 for O-DMAP; ENL:β=0.312, P<0.001 for EQP, β=0.210, P<0.001 for O-DMAP).

---

## [Decision Letter · Decision Letter 1]

31 Oct 2023

PONE-D-23-19702R1Lignan polyphenol metabolism is partially determined by isoflavone metabolism enterotypes: a cross-sectional analysisPLOS ONE

Dear Dr. Harada,

Thank you for submitting your manuscript to PLOS ONE. After careful consideration, we feel that it has merit but does not fully meet PLOS ONE’s publication criteria as it currently stands. Therefore, we invite you to submit a revised version of the manuscript that addresses the points raised during the review process.

We look forward to receiving your revised manuscript.

Kind regards,

Awatif Abid Al-Judaibi, PhD

Academic Editor

PLOS ONE

Journal Requirements:

Additional Editor Comments:

Dear authors,

Thank you for submitting your value work in PLOSONE, your positive response is appreciated.

I agree with reviewer2 that the title need improving and I am suggesting "A cross-sectional analysis of Lignan polyphenol partially determination by isoflavone metabolism enterotypes", or you can improve it by yourself, the word metabolism is written twice, and this may confuse the readers.

Reviewers' comments:

Reviewer's Responses to Questions

**Comments to the Author**

1. If the authors have adequately addressed your comments raised in a previous round of review and you feel that this manuscript is now acceptable for publication, you may indicate that here to bypass the “Comments to the Author” section, enter your conflict of interest statement in the “Confidential to Editor” section, and submit your "Accept" recommendation.

Reviewer #1: All comments have been addressed

Reviewer #2: All comments have been addressed

2. Is the manuscript technically sound, and do the data support the conclusions?

Reviewer #1: Yes

Reviewer #2: Partly

3. Has the statistical analysis been performed appropriately and rigorously? 

Reviewer #1: Yes

Reviewer #2: Yes

4. Have the authors made all data underlying the findings in their manuscript fully available?

Reviewer #1: Yes

Reviewer #2: Yes

5. Is the manuscript presented in an intelligible fashion and written in standard English?

Reviewer #1: Yes

Reviewer #2: No

6. Review Comments to the Author

Reviewer #1: The manuscript was corrected according to reviewers comments. Therefore, the manuscript may be accepted.

Reviewer #2: Dear Authors,

You've made good efforts to address the comments. I recommend a minor revision:

Make the title more catchy and concise. For instance, consider "Cross-Sectional Analysis of Lignan Polyphenol Metabolism and Isoflavone Enterotypes."

Ensure the conclusion is clear by summarizing the main findings, discussing their significance, addressing research questions and limitations, and providing data-based recommendations for future research.

Best regards.

7. PLOS authors have the option to publish the peer review history of their article (what does this mean?). If published, this will include your full peer review and any attached files.

Reviewer #1: No

Reviewer #2: **Yes: **Muhammad Zeeshan Ahmed

---

## [Author Response · Author response to Decision Letter 1]

15 Nov 2023

Responses for comments

Dear Editor and Reviewer,

Thank you for valuable feedbacks to our manuscript. We revised our manuscript based on your suggestions. Please see following responses and revisions.

Editor’s comment

Thank you for submitting your value work in PLOS ONE, your positive response is appreciated.

I agree with reviewer2 that the title need improving and I am suggesting "A cross-sectional analysis of Lignan polyphenol partially determination by isoflavone metabolism enterotypes", or you can improve it by yourself, the word metabolism is written twice, and this may confuse the readers.

(response)

Thank you for your comment.

As the Editor and Reviewer 2 suggested, we revised the title to “Association between lignan polyphenol bioavailability and enterotypes of isoflavone metabolism: a cross-sectional analysis” (L1–2 in changes-tracked manuscript).

In addition, we reviewed the manuscript and edits for typos.

Reviewer #1: The manuscript was corrected according to reviewers comments. Therefore, the manuscript may be accepted.

(response)

We appreciate your efforts in this review and evaluation of our study.

Reviewer #2: 

Dear Authors, You've made good efforts to address the comments. I recommend a minor revision:

#1: Make the title more catchy and concise. For instance, consider "Cross-Sectional Analysis of Lignan Polyphenol Metabolism and Isoflavone Enterotypes."

(response)

Thank you for your comment.

As the Editor and Reviewer 2 suggested, we revised the title to “Association between lignan polyphenol bioavailability and enterotypes of isoflavone metabolism: a cross-sectional analysis” (L1–2 in changes-tracked manuscript).

#2: Ensure the conclusion is clear by summarizing the main findings, discussing their significance, addressing research questions and limitations, and providing data-based recommendations for future research.

(response)

Thank you for the suggestion.

We revised the Conclusion section as suggested (L287–294 in the changes-tracked manuscript). 

In this study, the factors that are responsible for individual differences in the bioavailability of ENL and END were investigated in relation to enterotypes of isoflavone metabolism. Urinary END and ENL concentrations were 3.80- and 10.2-fold higher in the EQP/O-DMAP group than in the non-EQP/non-O-DMAP group, respectively. However urinary isoflavone concentrations were not associated with END and ENL concentrations. Dietary intake of lignan polyphenols may not represent precise bioavailable amounts and these enterotypes may provide better estimates. The possibility of biases and confounding cannot be ruled out due to the limitations in observed variables and participants. Further research to determine causative gut bacteria is needed.

Also, the values of lignans are provided (L156–160).

The geometric mean of the urinary log END (0.0473 μmol/g-Cr) value in non-O-DMAP/non-EQP was lower than for O-DMAP/EQP and O-DMAP/non-EQP (0.180 and 0.119 μmol/g-Cr; P < 0.001 and P = 0.0025, respectively). Also, urinary log ENL (0.0176 μmol/g-Cr) concentrations were lower in non-EQP/non-O-DMAP than in EQP/O-DMAP, non-EQP/O-DMAP, and non-O-DMAP/EQP (0.181, 0.0577 and 0.0943 μmol/g-Cr; P < 0.001, P < 0.001, and P = 0.0003, respectively).

---

## [Editor Report · Decision Letter 2]

16 Nov 2023

Association between lignan polyphenol bioavailability and enterotypes of isoflavone metabolism: a cross-sectional analysis

PONE-D-23-19702R2

Dear Dr. Kouji Harada,

We’re pleased to inform you that your manuscript has been judged scientifically suitable for publication and will be formally accepted for publication once it meets all outstanding technical requirements.

Kind regards,

Awatif Abid Al-Judaibi, PhD

Academic Editor

PLOS ONE

---

## [Editor Report · Acceptance letter]

24 Nov 2023

PONE-D-23-19702R2 

Association between lignan polyphenol bioavailability and enterotypes of isoflavone metabolism: a cross-sectional analysis 

Dear Dr. Harada:

I'm pleased to inform you that your manuscript has been deemed suitable for publication in PLOS ONE. Congratulations! Your manuscript is now with our production department. 

Kind regards, 

on behalf of

Professor Awatif Abid Al-Judaibi 

Academic Editor

PLOS ONE